# Making Molecules with Clay: Layered Double Hydroxides, Pentopyranose Nucleic Acids and the Origin of Life

**DOI:** 10.3390/life9010019

**Published:** 2019-02-15

**Authors:** Harold S. Bernhardt

**Affiliations:** Department of Chemistry, University of Otago, P.O. Box 56 Dunedin, New Zealand; harold.bernhardt@otago.ac.nz

**Keywords:** origin of life, layered double hydroxide (LDH) clay, pentose diphosphate, pentopyranose nucleic acid, arabinopyranose nucleic acid, RNA, base pairing, purine precursor, inosine, AICAR

## Abstract

A mixture of sugar diphosphates is produced in reactions between small aldehyde phosphates catalysed by layered double hydroxide (LDH) clays under plausibly prebiotic conditions. A subset of these, pentose diphosphates, constitute the backbone subunits of nucleic acids capable of base pairing, which is not the case for the other products of these LDH-catalysed reactions. Not only that, but to date no other polymer found capable of base pairing—and therefore information transfer—has a backbone for which its monomer subunits have a plausible prebiotic synthesis, including the ribose-5-phosphate backbone subunit of RNA. Pentose diphosphates comprise the backbone monomers of pentopyranose nucleic acids, some of the strongest base pairing systems so far discovered. We have previously proposed that the first base pairing interactions were between purine nucleobase precursors, and that these were weaker and less specific than standard purine-pyrimidine interactions. We now propose that the inherently stronger pairing of pentopyranose nucleic acids would have compensated for these weaker interactions, and produced an informational polymer capable of undergoing nonenzymatic replication. LDH clays might also have catalysed the synthesis of the purine nucleobase precursors, and the polymerization of pentopyranose nucleotide monomers into oligonucleotides, as well as the formation of the first lipid bilayers.

## 1. Introduction

Demonstrating the potential utility of a unified ‘systems chemistry’ approach to the problem of the origin of life, Powner and colleagues [1,2,3] recently proposed a link between a plausible early metabolic pathway and amino acid synthesis, through reactions between the small aldehyde phosphates glycolaldehyde phosphate (GAP) and glyceraldehyde-2-phosphate (G2P). Previously, Krishnamurthy and colleagues [4] showed that these same two molecules react—at millimolar concentrations, low-to-moderate temperatures and near-neutral pH—in the presence of layered double hydroxide (LDH) clays to produce pentose-2,4-diphosphates in 7% yield. One of these—ribose-2,4-diphosphate—constitutes the backbone subunit of pyranosyl-RNA [5], a pentopyranose nucleic acid that differs from RNA only in the position of its phosphate groups on the ribose ring, and in the ribose ring being six-membered (as opposed to five-membered as it is in RNA) (Figure 1). Pentopyranosyl nucleic acid systems have been found to constitute stronger base-pairing systems than RNA, with α-arabinopyranose nucleic acids possessing especially remarkable pairing strength [6]. In addition, the four pentopyranose phosphate systems with 4′→2′-backbone connectivity (including pyranosyl-RNA and α-arabinopyranose nucleic acid) are able to base pair with each other; in contrast, none of the four form base pairs with RNA. LDH-catalyzed reactions between GAP and G2P produce a mixture of different phosphorylated sugars, as shown in Figure 1, including tetrose-2,4-diphosphates (Figure 1A), pentose-2,4-diphosphates (Figure 1B, *boxed*) and hexose-2,4,6-triphosphates (Figure 1C) [4]; of these, the pentose-2,4-diphosphates are unique in constituting the backbone subunit monomers of oligonucleotides capable of base pairing [7].

Layered double hydroxide (LDH) clays are mixed-valence metal hydroxides with positively charged layers that adsorb charge-balancing anions such as Cl^–^, CO_3_^2–^ and PO_4_^3–^ within their aqueous interlayer [8,9]. These anions are exchangeable by diffusion with organic anions such as GAP and G2P, which bind through electrostatic interactions between the charged phosphate group and the charged layers, as well as through hydrogen bonding interactions with the LDH hydroxide groups. (A snapshot of a simulation of the interaction between a RNA 25-mer oligonucleotide and LDH [10] is shown in Figure 2). The distance between LDH sheets is remarkably flexible, with the LDH interlayer able to more than triple its width from ~7 Å to 24 Å in order to accommodate a polyanionic strand of DNA [11]. In catalysing the reaction between GAP and G2P, LDHs perform a number of key functions: 

Concentrating GAP and G2P from millimolar—or even micromolar—concentrations in the bulk medium to ~10 molar within the interlayer [4,12].

Binding GAP and G2P molecules in close proximity, thereby promoting their reaction.

Stabilizing and sequestering the reaction products, with 95 % of the pentose-2,4-diphosphate products still being extractable from the LDH after three months [4]. 

In these roles, the LDH clay is functioning as both a proto-enzyme and quasi-compartment, with Arrhenius [12] stating that, “Like cells, they retain phosphate-charged reactants against high concentration gradients and exchange matter with the surroundings by controlled diffusion through the ‘pores’ provided by the opening of the interlayers at the crystal edges” (p. 1580). LDHs also catalyse reactions between cyanide (CN^–^) anions to form diaminomaleonitrile, a precursor to the purine nucleobase adenine [13], and appear able to stabilize base pairs between RNA nucleotides in the adsorption of guanosine monophosphate (GMP) at low to moderate temperatures (Figure 3) [14]. Clay minerals such as LDHs are thought to have been present early in Earth’s history [15,16], with Arrhenius and colleagues suggesting that LDH clays such as hydrotalcite (a naturally-occurring Mg/Al-LDH clay) might have existed as “surface coatings on submerged weathering basalt, or in salt brine deposits in arid lakes” [17] p. 506, as occur on the Earth today. Another LDH mineral, green rust (an Fe^2+^/Fe^3+^-LDH), is thought to have been widespread on an early anoxic Earth, with the eventual rise in atmospheric oxygen causing its oxidation and precipitation from the ocean, giving rise to the ubiquitous banded iron formations. In relation to this, Russell has argued that, “the Hadean ocean crust…was likely thick, relatively cool, and covered in carbonate green rust” [18] p. 6.

Almost a decade ago, Powner and colleagues demonstrated a possible prebiotic synthesis of the pyrimidine nucleotides [19]; however, the search for a plausibly prebiotic synthesis of the purine nucleotides remains ongoing [20]. Kim and Benner have reported the synthesis of adenosine and inosine from reactions between ribose-1,2-cyclophosphate and the purine nucleobases [21], while Carell and colleagues have demonstrated synthesis of both ribopyranose and standard and non-standard RNA nucleotides starting with a derivatized pyrimidine as a molecular scaffold [22,23]. Powner and Szostak have also found a potential link with pyrimidine synthesis, discovering a branch point in Powner’s previously-discovered pyrimidine synthesis which leads to the non-standard 8-oxopurines, described as potential precursors to the standard purine nucleotides in the original report [24]. However, in subsequent work from the Szostak lab [25], the 8-oxopurines were found to be poor substrates in a nonenzymatic primer extension model system, which the authors now consider makes it unlikely that they played a role as purine precursors. They now believe that inosine—in contrast a good substrate in their model nonenzymatic system—is more likely to have played such a role [25]. We have previously proposed [26] that the prebiotic synthesis of RNA might have occurred through the progressive synthesis of the purine nucleobases on a pre-existing ribose-phosphate backbone, with the driving force for this synthesis being the increasing stabilization of the backbone through intermolecular interactions, including base pairing and—potentially—duplex formation. We also proposed that these nucleobase precursors were similar or identical to the intermediates of the modern *de novo* purine biosynthetic pathway, through which modern organisms synthesize purines [27], for example inosine and 5-aminoimidazole-4-carboxamide riboside (AICAR) (Figure 4). Furthermore, due to the plausibly prebiotic nature of many of the reactants in the modern purine biosynthetic pathway (such as glycine and CO_2_) [28], as well as the fact that two of the reactions also occur by alternative nonenzymatic reactions [29,30], we proposed that the biosynthetic pathway might have originated from a series of uncatalysed prebiotic reactions [26]. Could the first informational molecules have contained purines (and/or purine precursors) without pyrimidines? As discussed above, Szostak and colleagues have shown that the purine (precursor) inosine participates efficiently in nonenzymatic primer extension using a model system, leading them to conclude, in the words of their article title, “Inosine, but none of the 8-oxo-purines, is a plausible component of a primordial version of RNA.” [25] Crick [31] and Wachtershauser [32] have also previously argued in favour of an “all-purine precursor of nucleic acids”, although Wachtershauser has proposed a different sugar-nucleobase connectivity than exists in modern RNA.

## 2. Hypothesis

We propose that life has undergone (at least) two genetic transitions, the first from a mixed system of pentopyranose nucleic acids to RNA, and the second from RNA to a mixed RNA/DNA system. Because both transitions involved a *decrease* in the strength of base pairing, they must have been driven by selection for another property. RNA may have been selected for its catalytic ability. The relative flexibility of its backbone and consequent ability to adopt a variety of non-standard base-pairing interactions enables RNA to assume multiple complex 3D conformations, including binding motifs and catalytic sites [33,34]. In contrast, pentopyranose nucleic acids possess more rigid backbones and are relatively constrained in their base pairing interactions [7]; it therefore appears likely that these systems might possess a somewhat more modest catalytic repertoire than RNA. The expansion in the chemical landscape offered by RNA catalysis would see its ultimate achievement in the advent of coded peptide synthesis [35], which produced a virtual explosion in catalytic potential. The evolutionary driver from RNA to a mixed RNA/DNA system appears to have been DNA’s greater chemical stability, which allowed for long-term storage of genetic information as well as a massive increase in genome size [36]. Therefore, the selection criteria would have been different for each genetic transition. In contrast, initial selection of pentopyranose nucleic acids would have been due to the availability of the pentose-2,4-diphosphate backbone monomers through LDH-catalysed synthesis. In addition, pentopyranose nucleic acids appear able to undergo facile nonenzymatic replication, required prior to the advent of—RNA, or possibly pentopyranose nucleic acid—replicase enzymes able to catalyse these reactions. The large angle of inclination (~ 45º) between the base pairs and backbone in pentopyranose systems enables these oligonucleotides to undergo nonenzymatic replication using prebiotically-plausible 2′,3′-cyclic phosphate-activated monomers [7]. In contrast, in the case of RNA, the same 2′,3′-cyclic phosphate-activated monomers form unnatural 2′,5′-RNA phosphate linkages. It is unknown how a transition from a pentopyranose system to RNA would have occurred. It might have been directly, by—for example—the conversion of a six-membered ribopyranose ring to a five-membered RNA ring. However, if so, this would need to have occurred at the ribose phosphate level, as the transfer of the phosphate group from the 4′- to the 5′-OH first requires opening of the six-membered pyranose ring, which would have been prohibited at the nucleotide stage by the attachment of the nucleobase (precursor). 

In addition to catalysing the synthesis of the monomer units of the pentopyranose–phosphate backbone, we propose that LDH clays played a role in the synthesis of the purine nucleobase precursors, either by catalysing reactions between cyanide anions [13] or through the promotion of reactions between prebiotically plausible molecules such as glycine, NH_3_ and CO_2_, similar to those which occur in the modern *de novo* purine biosynthetic pathway [26]. Greenwell and colleagues have demonstrated peptide bond synthesis catalysed by a ternary Mg/Cu/Al-LDH clay under wet–dry cycles [37], in an analogous reaction to the amide bond formation that occurs in the biosynthesis of GAR, the first stable intermediate in the purine biosynthetic pathway [26,27]. Krishnamurthy and colleagues have shown that LDH clays catalyse the formylation of GAP utilizing a sulfite–formaldehyde adduct [38], in a parallel to the two indirect formylation reactions that occur in the purine biosynthetic pathway. As described above, Gwak and colleagues [14] have produced evidence that GMP nucleotides in the presence of LDHs form non-standard base pairs at low to moderate temperatures (20–60 °C), whereas at higher temperatures (80–100 °C) only unpaired GMP is present (Figure 3). This suggests that the LDH-promoted polymerisation of pentopyranose nucleotide monomers to form oligomers might also be possible. In addition, montmorillonite, a phyllosilicate clay containing negatively charged sheets that bind Mg^2+^ and other cations, catalyses the polymerisation of RNA nucleotides to form up to 50-mer RNA oligos [39], suggesting such reactions might also be possible for LDH clays. Supporting this possibility, RNA nucleotides [14,40,41], oligonucleotides and duplexes [42] are strongly adsorbed by LDH clays through their charged phosphate backbones, as demonstrated both experimentally and in simulation experiments (Figure 2). Polymerisation of phosphorylated sugars lacking an attached nucleobase (precursor) may also be possible, as demonstrated by the putative polymerisation of fructose-1,6-bisphosphate (a sugar diphosphate with a similar structure to a pentose-2,4-diphosphate), in the presence of a Li^+^-LDH [43]. The high affinity of LDH clays for phosphate anions is rather striking, and suggests that these clays might have played a critical role in extracting and concentrating phosphate from low background levels in the prebiotic environment; however, this raises the question of how the strongly bound pentopyranose oligonucleotides might have been released from the LDH interlayer. Russell has posited that LDH clays (including green rust) initially formed in the high pH environment of alkaline deep-sea hydrothermal vents [18], which suggests the possibility that release from the LDH interlayer might have occurred in the increasingly acidic conditions distal from the vent, as LDH clays largely disintegrate at acidic pH [8]. Exchange of the phosphate moieties with divalent carbonate anions [8], possibly through the interaction with high atmospheric levels of CO_2_, would have been another possible mechanism for the eventual release of pentopyranose oligonucleotides from their LDH ‘cells’. 

Conceivably, the high pairing strength of pentopyranose systems might be considered a disadvantage for nonenzymatic replication, as it could increase the chances of mispairing and make the system vulnerable to product inhibition (wherein the just-synthesized copy remains bound to the template sequence, preventing further replication) [44]. However, as described above, we previously proposed that the first nucleobases were precursors to the purine nucleobases [26]—perhaps the same or similar to the intermediates of the modern *de novo* purine biosynthetic pathway—and it would seem reasonable that weaker interactions between the precursors might have offset this high pairing strength. Two examples of possible precursors are inosine and AICAR (Figure 4). Inosine is a purine itself, as well as a precursor to adenosine and guanosine, and in fact plays a key role in genetic coding, occurring in the anticodon wobble position of alanine tRNA [45]. Interestingly, it is utilised due to its ability to form a *purine–purine base pair* (with adenosine) as well as more typical purine–pyrimidine base pairs with cytosine and uracil. When incorporated into RNA oligonucleotides, inosine and 1-(2-deoxy-β-d-ribofuranosyl)-imidazole-4-carboxamide (dICAR) (a 2′-deoxy analogue of AICAR) form base pair interactions with purines as well as pyrimidines [46,47,48]. The similarity in hydrogen bond-forming potential between adenosine, AICAR and dICAR is shown in Figure 4. However, these pairings between purine precursors are weaker—in some cases significantly—than standard purine–pyrimidine base pairs. In the case of dICAR, this is presumably due to a decrease in base-stacking stabilization due to AICAR possessing an imidazole ring as opposed to a purine double ring structure. Self-complementary RNA oligonucleotide 12-mers containing two inosine- or dICAR base pairs have melting temperatures ~20 °C lower than the comparable RNA duplexes containing standard purine–pyrimidine base pairs only [47], and it is likely that the presence of inosine and AICAR would similarly decrease the stability of pentopyranose duplexes, although this has not been shown experimentally. Nevertheless, we propose that this decrease in stability would have been offset by the greater strength of pentopyranosyl pairing interactions. In addition, it has been shown that pentopyranose duplexes accommodate purine–purine base pairs more easily than RNA duplexes [7], which a number of these interactions would constitute (or at least approximate). In fact, the two opposing effects—inherently more stable duplexes vs. weaker interactions between purine precursors—might have produced a system able to undergo nonenzymatic replication: strong enough for duplex formation and replication, but not so strong as to produce template-product inhibition [44]. 

The lack of cross pairing between pentopyranose and RNA systems—and the consequent inability to transfer sequence information in the proposed genetic transition—could be seen as a major weakness of our hypothesis. However, genetic transitions will as a rule result in a degree of information loss, due to the alteration in 3D structure (and therefore function) caused by the difference in backbone conformation between the preceding and succeeding informational polymers [7]. As Eschenmoser has pointed out, “Irrespective of its direct communication with the predecessor system, the successor system would have to evolve its phenotype (i.e., chemical catalytic functionality] *de novo*.” [7] If, however—as is the case with the non-cross pairing pentopyranose and RNA systems—there is *zero* possibility of information transfer, one might ask what is the point of proposing such a transition, and the need for a pentopyranose system to evolve first? Wouldn’t the principle of Occam’s razor favour the hypothesis that RNA arose *de novo?* As discussed above, advantages of the hypothesis are:The demonstrated prebiotically plausible formation of the pentose–diphosphate backbone monomers; and, conversely, the absence of this for the RNA backbone subunit. The structural features of pentopyranose systems, for example the large angle of inclination between base pairs and backbone, that make them structurally suited for nonenzymatic replication using plausibly prebiotic 2′,3′-cyclic phosphate-activated monomers. The greater pairing strength of pentopyranose systems—important to counterbalance the initial weaker interactions between purine nucleobase precursors—to allow stable duplex formation and nonenzymatic replication.The fact that a pentopyranose system provides a—structural and possibly also catalytic—stepping stone to RNA, removing some of the difficulties in the RNA backbone arising *de novo.*

Finally, it seems likely that LDH clays might also have played other roles in the origin of life, such as in the formation of the first lipid bilayers. The negatively charged carboxylate anions of CH_3_(CH_2_)_16_COO^–^Na^+^, the sodium salt of the long-chain fatty acid octadecanoic (or stearic) acid, adsorb to both sides of the LDH interlayer, forming a tilted version of a lipid bilayer, and increasing the interlayer width to 48 Å (Figure 5) [49]. In conclusion, what we have learned so far regarding LDH clays points to our having only just begun to scratch the surface in relation to this fascinating class of materials.

## Figures and Tables

**Figure 1 life-09-00019-f001:**
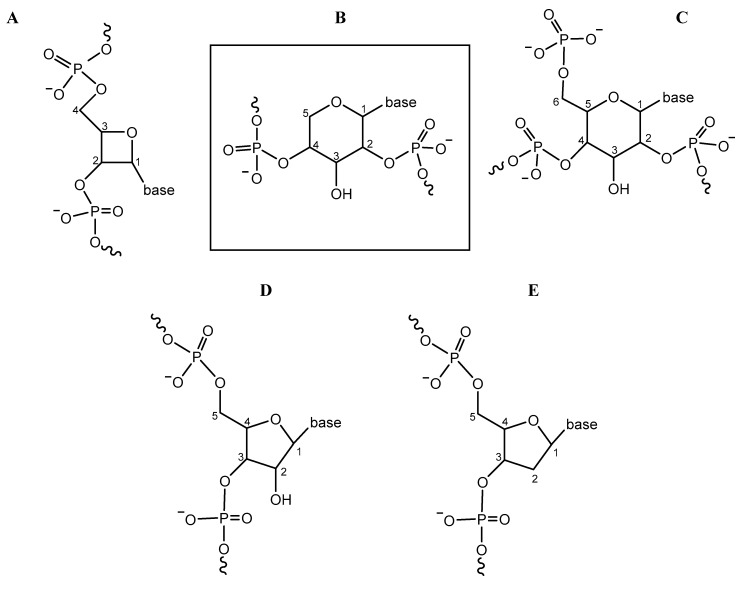
Nucleic acid backbone subunits, showing position of phosphate groups and attached (nucleo)base. (**A**–**C**) (minus base) are produced in reactions catalysed by layered double hydroxide (LDH) clay, as described in the text; of these, only (**B**) (boxed) (present in pentopyranose nucleic acids) is capable of base pairing. In contrast, the backbone subunits of RNA (**D**) and DNA (**E**) do not have a plausibly prebiotic synthesis. A = tetrooxetose nucleic acid, B = pentopyranose nucleic acid, C = hexopyranose nucleic acid, D = RNA, E = DNA. Numbering of carbohydrate carbon atoms is shown. Figure produced in ChemDraw^®^ 18, PerkinElmer Informatics, Waltham, MA.

**Figure 2 life-09-00019-f002:**
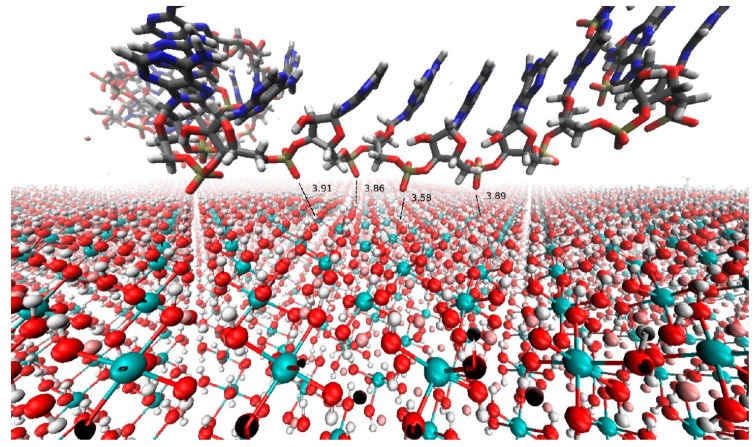
Computer simulation snapshot of the interaction between an RNA 25-mer and LDH interlayer surface, depicting the inner-sphere complexes between RNA phosphate oxygen atoms (red) and LDH metal hydroxide hydrogen atoms (white) in the absence of bridging water molecules. Numerical values shown in black indicate the distance between interaction donors and acceptors in angstroms. Color scheme: O (red), H (white), P (gold), C (grey), N (dark blue), Mg (cyan) and Al (pink). Reprinted with permission from [10]. Copyright 2013 American Chemical Society.

**Figure 3 life-09-00019-f003:**
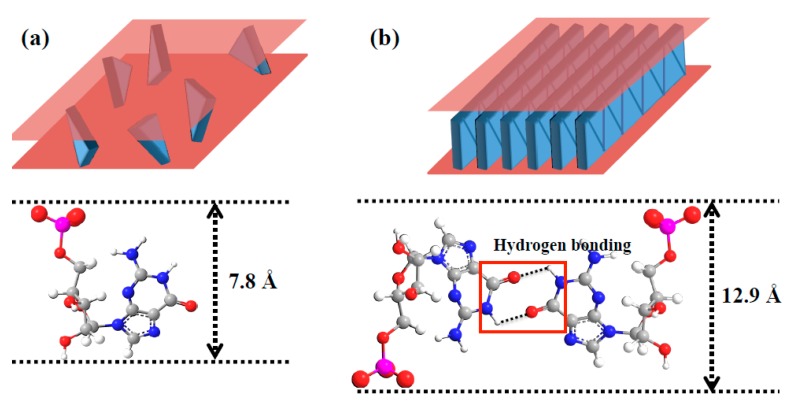
Schematic illustrations for interlayer structure of GMP/LDH hybrids according to molecular arrangement of GMPs: (**a**) single molecule arrangement (GL-S) and (**b**) ribbon II arrangement (GL-R). GL-S: 12.6 Å (d-spacing) − 4.8 Å (LDH layer thickness) = 7.8 Å; GL-R: 17.7 Å (d-spacing) − 4.8 Å (LDH layer thickness) = 12.9 Å. Adapted with permission from [14].

**Figure 4 life-09-00019-f004:**
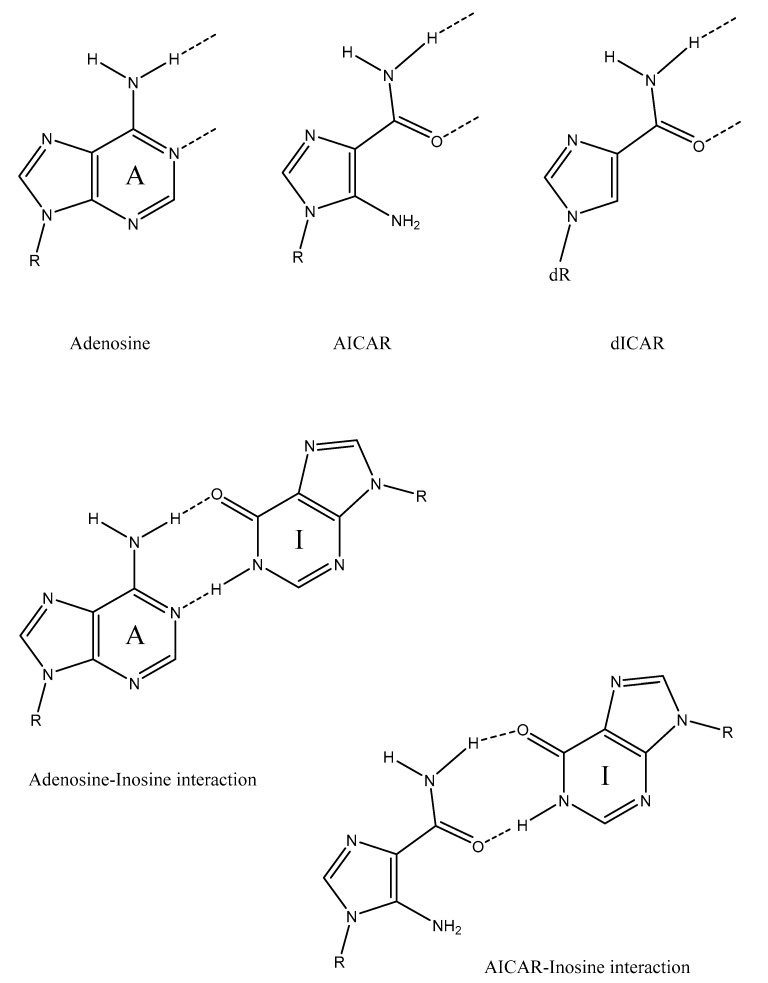
(**Top**) the potential of 5-aminoimidazole-4-carboxamide riboside (AICAR) (an intermediate in the modern *de novo* purine biosynthetic pathway) and its 2′-deoxy analogue 1-(2-deoxy-β-d-ribofuranosyl)-imidazole-4-carboxamide (dICAR) to form hydrogen-bonding interactions similar to those formed by adenosine. (**Bottom**) hydrogen-bonding (base pairing) interaction between adenosine and inosine (a later intermediate in the purine biosynthetic pathway), and proposed interaction between AICAR and inosine, examples of the weaker interactions between purine nucleobase precursors proposed to have preceded purine-pyrimidine base pairing. R = ribofuranose; dR = 2′-deoxyribofuranose, figure produced in ChemDraw^®^ 18 (PerkinElmer Informatics, Waltham, MA).

**Figure 5 life-09-00019-f005:**
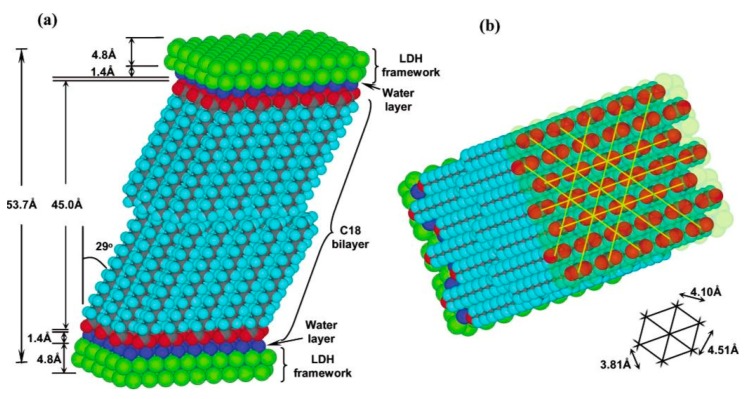
Structural models of the stearate/LDH interaction showing (**a**) tilted stearate bilayer and (**b**) regular packing of stearate anions in the interlayer gallery. Reprinted with permission from [49]. Copyright 2003 American Chemical Society.

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
