# Peer review of "Making Molecules with Clay: Layered Double Hydroxides, Pentopyranose Nucleic Acids and the Origin of Life"

_life, 2019, doi:10.3390/life9010019_

Round 1
Reviewer 1 Report
The paper deals with an interesting topic, the origins of nucleic acids, building a theoretical argument on Eschenmoser's reported synthesis pRNA's sugar and reported oligomerisation of pRNA cyclic phosphates.
However, the author make a number of fundamental assumptions that are not supported by any evidence and these undermine the whole paper. These issues need to be addressed.
1. For example, in Figure 2. Pyranosyl and furanosyl nucleotides are not in equilibrium and it is not reasonable to assume that "transfer of phosphate group(s) on a ribopyranosyl monomer, [would] convert the six-membered ribose ring of a ribopyranosyl nucleotide into the five-membered ribose ring of RNA nucleotides." There is no evidence this can be achieved. By what mechanism can this interconversion take place?
2. The author must cite recently reported synthesis of ribonucleotides, such as by Sutherland and coworkers.
3. The author must consider how the sugar phosphates Eschenmoser reported might be turned into nucleotides? These reactions are not known, and without them these arguments are futile.
4. pRNA and RNA have different helical twists and don't communicate (base pairs) together, therefore information transfer and genetic takeover is very unlikely. If all information is lost during the transition; what is the benefit of the early stage? Why not just directly make RNA if there is no information transfer from pRNA to RNA?
Author Response
Please find my detailed response to reviewer 1 attached.

Reviewer 2 Report
Comments over the article in the PDF attached.
Major concerns are in the grey background.

Author Response
Please find my detailed response to Reviewer 2 attached.

Reviewer 3 Report
The hypothesis article by Harold S. Bernhardt describes the role of pentose-2,4-diphosphates as monomers, and pentopyranosyl nucleic acids, which constitute stronger base-pairing systems than RNA as the backbone in the evolution of life on earth ad argues that pentopyranosyl nucleic acids were the first informational molecules to arise on the Earth. The role of LDH clays in the development of these material also described.
Eventhough the article is highlighting the aspects of LDH clays in the formation of pentose-2,4-diphosphates, I feel the article is limited to one system, that is pentopyranosyl nucleic acids. If the author has to emphasize the role of LDH clays/minerals in the origin of life, I feel more examples of using such systems should de described along with pentopyranosyl nucleic acids. Eventhough the title of the article is “Making molecules with clay”, the article is limited to one example. The role of LDH clays in the development of other prebiotic materials, if any, could be a good addition to this paper and that will improve the scope of this publication.
In other way, if the author would like to emphasize the pentopyranosyl nucleic acids as first informational polymer in the origin f life, I feel, the article should be strengthened with reported literature and make the paper more interesting by incorporating results, graphics and conclusions from other reports in this particular linkage.
Otherwise, the author should modify the abstract by adding expansion of GAP, G2P as it is difficult for a general reader to understand these terms from the beginning.
The author wrote that “Consequently, pentopyranosyl nucleic acids may have been both what ‘stuck’ to these clays 97 most strongly, and what stuck the clays together, protecting against their dissolution. As for getting 98 ‘unstuck’, LDH clays become solubilized at acidic pH, and so a drop in pH would have released the 99 pentopyranosyl strands from their LDH sheet ‘moorings’.”. But it is not clear how this will happen and what is the mechanism of action. The author should explain this with proper citations.
I think, the chemical structures could be modified by following a format, may be ACS format to make them beautiful.
It is wrote that “Figure 3B shows a possible base pair-like interaction between 148 AICAR and inosine modelled on an adenosine–inosine base pair”. I will recommend the author to explain the details of this modeling with proper citations.
The author should correct all references to make them uniform by incorporating the DOI numbers to all.
Author Response
Please find my detailed response to Reviewer 3 attached.

Round 2
Reviewer 1 Report
The authors have made good improvements to the previous manuscript.
Figure one would be improved with the addition of stereochemistry.
Author Response
Thank you to the reviewer for their positive comments regarding the revised manuscript. As is often the case, I believe the comments and criticisms of the three reviewers in their First Round Reports has resulted in a much-improved MS.
Regarding the lack of stereochemistry in Figure 1, this was something that I had thought of. However, the first reference to the figure relates to the different ring sizes, and different positions of the phosphate groups in the pentopyranose (Fig. 1B) vs. RNA (Fig. 1D) backbones, and I believe this is much clearer and easier to see in the 2D representations (along with the numbered carbon atoms). Also – and more importantly – the three types of backbones (Fig. 1A, B and C) produced in the reactions catalysed by LDHs include multiple isomers, and so would require a much larger figure to properly represent all the stereochemistries produced. For example, there are four pentopyranose nucleic acids all with different stereochemistries: ß-ribopyranose, arabinopyranose, xylopyranose and lyxopyranose. The tetrooxetose (Fig. 1A) and hexopyranose (Fig. 1C) backbones likewise have multiple stereoisomers. For all the above reasons, I believe the current figure is the best option.
Reviewer 2 Report
The manuscript has improved significantly upon revision. Author has put significant work in this revision, and I would like to recommend the manuscript to publication.
Author Response
Thank you to the reviewer for their positive comments regarding the revised manuscript. As is often the case, I believe the comments and criticisms of the three reviewers in their First Round Reports has resulted in a much-improved MS.
Reviewer 3 Report
The manuscript can be accepted in the present form
Author Response
Thank you to the reviewer. As is often the case, I believe the
comments and criticisms of the three reviewers in their First Round
Reports has resulted in a much-improved MS.